# Computational multiplex panel reduction to maximize information retention in breast cancer tissue microarrays

**Luke Ternes**[1], **Jia-Ren Lin**[2], **Yu-An Chen**[2], **Joe W. Gray**[1], **Young Hwan Chang**[1]*

**1** Department of Biomedical Engineering, Oregon Health and Science University, Portland, Oregon, United States of America, **2** Ludwig Center at Harvard and Laboratory of Systems Pharmacology, Harvard Medical School, Boston, Massachusetts, United States of America

* chanyo@ohsu.edu

**Data Availability Statement:** Code available from the author's Github and tutorials (https://github.com/GelatinFrogs/ME-VAE_Architecture) and

## Abstract

Recent state-of-the-art multiplex imaging techniques have expanded the depth of information that can be captured within a single tissue sample by allowing for panels with dozens of markers. Despite this increase in capacity, space on the panel is still limited due to technical artifacts, tissue loss, and long imaging acquisition time. As such, selecting which markers to include on a panel is important, since removing important markers will result in a loss of biologically relevant information, but identifying redundant markers will provide a room for other markers. To address this, we propose computational approaches to determine the amount of shared information between markers and select an optimally reduced panel that captures maximum amount of information with the fewest markers. Here we examine several panel selection approaches and evaluate them based on their ability to reconstruct the full panel images and information within breast cancer tissue microarray datasets using cyclic immunofluorescence as a proof of concept. We show that all methods perform adequately and can re-capture cell types using only 18 of 25 markers (72% of the original panel size). The correlation-based selection methods achieved the best single-cell marker mean intensity predictions with a Spearman correlation of 0.90 with the reduced panel. Using the proposed methods shown here, it is possible for researchers to design more efficient multiplex imaging panels that maximize the amount of information retained with the limited number of markers with respect to certain evaluation metrics and architecture biases.

## Author summary

Multiplex tissue imaging techniques utilize large panels of markers that attempt to gather as much information as possible, but increasing the number of stains does come with the downsides of increased autofluorescence and tissue degradation. There exists a theoretical subsampling of markers that is able to recreate the same information as a full panel; therefore, removing the self-correlating information with such a subset would increase the efficiency of the imaging process and maximize the information collected. By selecting an idealized subsample of markers, a deep learning model can be trained to predict the same

(https://github.com/GelatinFrogs/
PanelSelectionMethods).

**Funding:** This work was supported in part by the
National Cancer Institute – U54CA209988 (J.W.G),
U2CCA233280 (J.W.G), U2C-CA233262 (P.K.S),
R01 CA253860 (Y.H.C) and Kuni Foundation
Imagination Grants (Y.H.C). The funders had no
role in study design, data collection and analysis,
decision to publish, or preparation of the
manuscript.

**Competing interests:** I have read the journal's
policy and the authors of this manuscript have the
following competing interests: L.T., J.R.L., Y.C and
Y.H.C have no competing interests. J.W.G has
licensed technologies to Abbott Diagnostics; has
ownership positions in Convergent Genomics,
Health Technology Innovations, Zorro Bio, and
PDX Pharmaceuticals; serves as a paid consultant
to New Leaf Ventures; has received research
support from Thermo Fisher Scientific (formerly
FEI), Zeiss, Miltenyi Biotech, Quantitative Imaging,
Health Technology Innovations, and Micron
Technologies.

information as a full dataset with fewer rounds of staining. Here we evaluate several methods of subsample marker selection and demonstrate their ability to reconstruct the full panel's information.

This is a *PLOS Computational Biology* Methods paper.

## Introduction

The imaging of histologic features and the examination of cellular subtypings are essential components of modern-day cancer research and clinical management [1–6]. As such, many state-of-the-art multiplex imaging modalities, such as Cyclic Immunofluorescence (CyCIF) [7], co-detection by indexing (CODEX) [8] and multiplexed immunohistochemistry (mIHC) [1], have been developed that allow for a more in-depth interrogation of spatial biology by staining a single tissue sample with dozens of markers. The use of these modalities and vast panels have provided an unprecedented tool for tissue spatial biology and enabled better understanding into aspects of cell phenotyping and spatial distribution [3,6,9,10]. These imaging modalities come with their own downsides, however, since they require specific expertise necessitating choosing panels and the order of staining, and resources as they can take $\sim$ 1 week or more to collect data depending on tissue and panel size [7].

Despite the increased potential for high-dimensional single cell phenotyping that may serve as diagnostic biomarkers or therapeutic targets, the panels used for an experiment must still be chosen intelligently and with purpose because each additional round of staining comes with increasing levels of autofluorescence, tissue degradation, and tissue loss that can make later rounds of staining unusable for downstream analysis [7,11]. As such, when selecting a panel, experts must pay considerable attention to the biology of the disease in order to capture specific features of interest, but it is not always possible for experts to know the full extent of marker co-expression, co-localization, and importance within a dataset ahead of time. Data generation and analysis are expensive and time consuming, and choices often have to be made to reduce the scope of experiments, which will result in the loss of potentially vital information. If we can computationally determine the extent of marker co-expression and predictability beforehand, then we can reduce the panel sizes of breast cancer (BC) tissue microarrays (TMAs) in such a way that the maximum amount of information is retained in a minimum image size with the optimized marker panels.

Previous methods attempting to reduce the burden and expense of immunofluorescence imaging have sought to use the information found in Hematoxylin and Eosin (H&E) images alone [12,13], transmitted light microscope images [14], and electron micrograph inputs [15] to predict immunofluorescence staining information with decent success. We reasoned that just as it is possible to use deep learning architectures to predict immunofluorescence information from features in H&E or another modality alone, so too is it possible to predict shared information from one marker or combination of markers to another within a breast cancer CyCIF dataset. Markers with high degrees of co-expression can easily be imputed from the expression of other similar markers on the panel and therefore can be removed, while other markers might have no co-expressed marker pairs and therefore cannot be removed. By selecting an idealized subsample of markers, a deep learning model such as variational autoencoder (VAE) can be trained to predict the same information as a full dataset with fewer rounds of staining.

In this study, we propose a two-steps approach: 1) create optimally-designed reduced marker panels and 2) reconstruct the full panel's information from a reduced marker set. We evaluate several methods of optimally reduced marker selection and demonstrate the ability of deep learning architectures to reconstruct the full panel's information from a reduced marker set using BC TMA data with CyCIF staining.

## Results

### Proof-of-concept for using a generative model to impute missing markers

Image-to-Image translation is the task of taking images from one domain and transforming them into another domain by learning the mapping between two domains [16], and it can be applied to a wide range of applications including biomedical imaging [12–15,17,18]. Also, recent studies showed theoretically and experimentally, when genes are co-regulated, one might infer the expression of other individual genes from a much smaller number of composite measurements [19,20]. Thus, we reasoned that it is possible to predict the CyCIF staining of one panel using another panel or combination of stains. This can be used to reduce the number of stains in CyCIF protocols by using a reduced panel set to predict a larger panel of markers without actually having to stain for them. The question then becomes, what is the theoretically best selection of markers for maximizing the amount of information retained and generating the whole panel image predictions.

To test this process, we use a breast cancer tissue microarray (TMA) dataset comprised of 88 cores, 6 different breast cancer subtypes (plus normal), and 25 markers in the CyCIF panel (S1 Table) (see Methods and Supplementary Information). The panel consisted of both general cell structure markers such as Hoechst and Lamin for nuclear staining, as well as specific breast cancer subtype markers such as multiple cytokeratins, estrogen receptor, progesterone receptor, and human epidermal growth factor receptor 2 (HER2). The full panel also includes immune markers such as CD3 and CD45, vascular markers such as CD31, and cancer markers such as E-cadherin. This panel was designed by researchers for other biological studies and is therefore reflective of real-world experiments and data. Our implementation consists of three steps as shown in Fig 1: A) panel selection, B) train deep learning model to reconstruct full panel information, and C) evaluation. For selecting an optimal reduced panel set, we evaluate 4 different methods: correlation-based selection, sparse subspace-based selection, gradient-based selection, and random selection as shown in Fig 1 (see Methods). The reduced panels of each method were then used to reconstruct the initial full panel using a multi-encoder

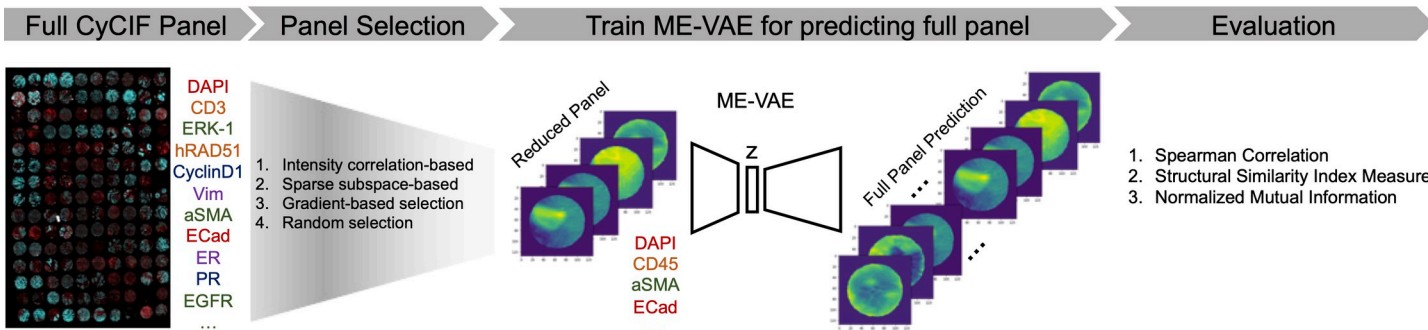

**Fig 1. An illustrative schema for panel reduction and prediction: In order to select an optimally reduced panel from a designed full panel, four different selection methods were tested: intensity correlation-based, sparse subspace-based, gradient-based, and random selection.** Using the reduced panels selected from each method, a ME-VAE was used to impute the full panel set. The full set of imputations were then evaluated by comparing them to the original images using important features of downstream analyses such as mean intensity correlations, structural similarity index measure, and cluster overlap using Normalized Mutual Information.

variational autoencoder (ME-VAE) [9], which encodes the markers of single cell image in the reduced panel into a latent descriptor and generates all 25 markers of single cell image in the full panel set. The reconstructed images of each method are then evaluated using various metrics including single-cell based structural similarity index measure (SSIM) from the reconstructed image, mean intensity correlation between real stained and the reconstructed image, and cluster overlap to determine whether information is retained in the reduced panel and prediction pipeline.

Before we create optimally-designed reduced marker panels, we evaluate a proof of concept to demonstrate how information from a reduced panel set can adequately predict unseen information within the full set using ME-VAE model. To do this, we first randomly selected 50% of the full panel as shown in (S2 Table) which was used to predict the other 50% marker panel as shown in Fig 2. As can be seen qualitatively in the real and predicted image pairs, the morpho-spatial features of size, shape, distribution, and relative intensity are preserved, regardless of whether the marker was present in the training or withheld panel. Quantitatively this is captured using the structural similarity index measure (SSIM)[21], which is a widely used measure of image similarity as perceived by the human visual system[12]. The overall quantification shows a mean SSIM of 0.75. The predictions also achieve a Spearman correlation 0.75 to the withheld stains, which demonstrate feasibility of marker prediction using a generative model with reduced panel.

To help ground the quantitative metrics in this proof-of-concept experiment in a more interpretable context, we compare the degree of error to other forms of common technical noise (blurring, salt/pepper, and differences in segmentation as simulated by erosion/dilation)

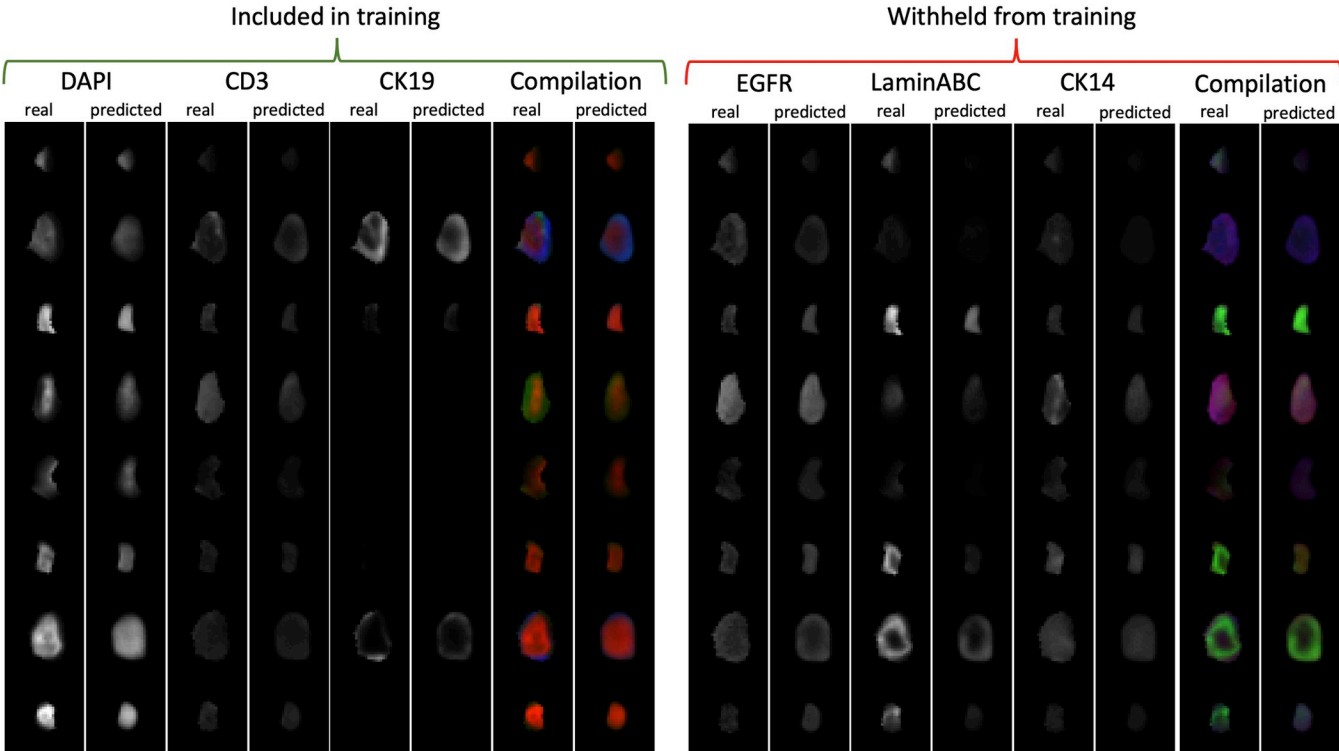

**Fig 2. A proof of concept full panel imputation: 12 stains were randomly selected to create a reduced panel which were then used to train a ME-VAE to reconstruct the full panel of stains.** Here we show a representative 3 stains from the included and withheld marker sets across 8 cells. Real and predicted staining and their compilation images are shown side by side to qualitatively demonstrate that a reduced panel can reconstruct relevant unseen information.

as can be seen in S1 Fig. One can see visually and quantitatively that each of the different noises have varying degrees of severity, with blurring having the smallest effect on intensity and overall structure of the image. Differences in segmentation via erosion/dilation has the largest effect as the inclusion and exclusion of a few pixels can make a significant difference when the overall size of the cell is only about 10–20 pixels across [12]. The effect of this mis-segmentation will also have a larger effect on membrane stains or densely packed cell populations. With regards to the structure of the predicted image, the randomly selected panel of 12 stains achieves an average SSIM of 0.75, just behind blurring at 0.78 and well above salt/pepper and erosion/dilation at 0.68 and 0.67, respectively. Although the predictions from the randomly chosen panel of 12 stains rank lowest in mean intensity correlation at 0.80, the score is still comparable to segmentation noise at 0.83. Framed in this context one can see that even when a panel is randomly reduced by 50%, it is possible to recapitulate the full spectrum of information held within the full panel within the margins of normal technical variation. Technical noise and deviance like those shown in S1 Fig are common and accepted in biomedical image analysis. Although perfect information retention would be ideal, the reconstructions from the reduced panel are shown to be comparable to other forms of accepted noise and variance. Evaluation of the utility of these reconstructions will be discussed in later experiments. The question still remains, however, to what degree selecting the reduced panel methodologically can improve these results. Thus, we further focus on how to optimally select a reduced marker panel, instead of testing several reconstruction architectures. To do this, each selection method is given the same full panel information to select from, and then we evaluate its selection using the same reconstruction architecture across all selection methods for comparable evaluation of reduced panels.

## Evaluating panel reduction methods with imputed marker correlation

Our evaluation of panel reduction methods was conducted by correlating the original and reconstructed panels' mean intensities at the single cell level as shown in Fig 3 and quantified in S3 Table. This was done over an increasing number of stains in the reduced set to show how each method performs with different levels of information available in the reduced panel.

For a baseline comparison, the intensities of the reduced panel were used as a 1-to-1 substitute for the missing stains; for example, if CK19 is included in the reduced panel and PanCK is not included, then the CK19 expression will be directly used as the prediction of PanCK expression since it is PanCK's highest correlate within the reduced panel. This baseline of 1-to-1 substitution resulted in subpar predictions that did not converge to a correlation approximating the full panel until nearly all stains were included in the reduced set, indicating the need for predictive models to retain information on removed markers (Fig 3).

Random selection performed moderately better than baseline, achieving a mean Spearman correlation of 0.77 for withheld markers and 0.89 for all markers when 18 of 25 markers (72%) are included in training. Additionally, random selection had a variance in its correlations of 0.005 in both the withheld set and full set. As previously mentioned in the proof-of-concept above and further seen here, the random selection of a reduced panel performs well. This, however, is primarily a result of the generative capacities of deep learning models to process and predict patterns missing from the data. Random selection here can be used as a computational baseline to illustrate the increased performance and predictive power that comes from deep learning regardless of intelligent panel design. Without the constraints of time and processing power, this study could be improved by training and evaluating predictive models using dozens of randomly selected panels; however, each deep learning model (for each panel arrangement and size) can take more than a day to train and evaluate. For this study only a

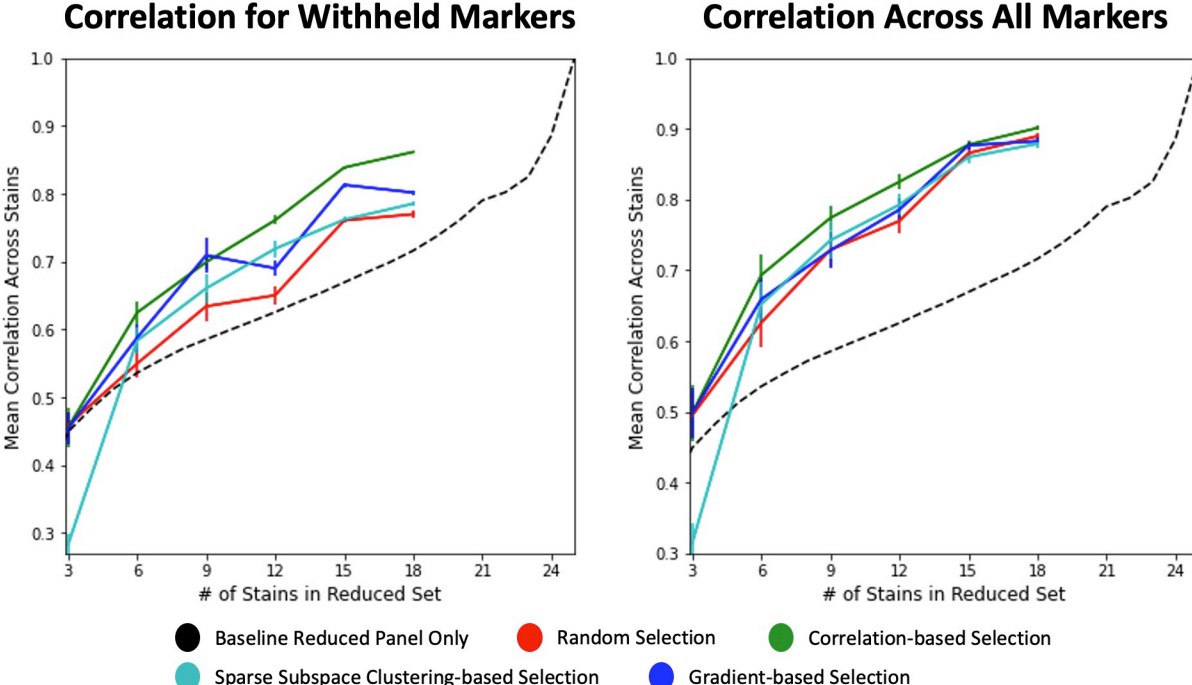

**Fig 3. All panel selection methods were evaluated across a range of panel sizes to determine how well their reduced panels can be used to reconstruct the full panel.** Spearman correlation was measured for each stain independently and then averaged across the whole dataset. Variance in mean correlation accuracy was also calculated across markers for each panel size and method (n = 25). The data was split into withheld markers (left) and all markers (right) to illustrate each model's generalizability and performance in both domains. 1-to-1 substitutions of marker intensity were used as the baseline, where makers withheld from the reduced panel set were simply assigned the intensity of their closest match as described in the Methods section.

single randomly selected panel at 6 different sizes is used, which prohibits analysis as to the variability of random predictions.

Sparse subspace-based selection performs slightly better than random selection and baseline, achieving mean Spearman correlations of 0.80 and 0.89 for withheld markers and all markers, respectively. Subspace-based selection had variances in its correlations of 0.004 in the withheld set and 0.005 in full set. By looking at the correlations with respect to panel size (Fig 3), however, one can see that sparse subspace-based selection performs even better at lower panels sizes compared to random, for instance, correlation for withheld markers with 12 stains in reduced set.

Gradient-based selection performs better than random or sparse subspace-based selection methods within the withheld marker predictions, achieving a max correlation of 0.81. It is worth noting that this prediction method appears to be less stable with prediction metrics fluctuating when different panel sizes are used. Within the full marker set, gradient-based selection performs similarly to random and subspace selection methods, achieving a correlation of 0.88. Gradient-based selection had variances in its correlations of 0.003 in the withheld set and 0.006 in full set.

Finally, correlation-based selection also performs well at reconstructing the mean intensities of each stain, both within and withheld from the reduced panel. For most every panel size, correlation-based selection achieves the highest Spearman correlations compared to the other selection methods, obtaining a correlation of 0.86 and 0.90 for withheld and all markers, respectively. For the purpose of reconstructing mean intensity information, the other selection methods only perform similarly to correlation-based selection at extremely low panel sizes

where there is insufficient information. Gradient selection had variances in its correlations of 0.001 in the withheld set and 0.003 in full set.

## Model generalizability

In order for a reduced panel to be consistently effective, it must be generalizable across datasets and similar pathological states. An example of this is breast cancer subtyping where unique expression patterns will vary and classification can fail if important markers are not able to be predicted properly based on the underlying information of that specific biology. To test this pathological state generalizability, we applied the highest performing panel (correlation-based with 18 markers) to all the different cancer subtypes within the BC TMA dataset composed of 88 cores and different BC subtypes including luminal A, luminal B, luminal B/HER2, HER2, Triple Negative (TN), Invasive Lobular Carcinoma (ILC), Normal Breast separately and evaluated the predicted expressions as shown in Fig 4. Although there is some slight variance, the panel performs well consistently across all subtypes, achieving Spearman correlations between 0.72 and 0.83 within the excluded markers. However, the specific markers that scored the highest and lowest correlations in each subtype, did vary based on the relative biological expression. It can be further observed in Fig 5 that the markers that performed poorly simply did not have large variation across the specific subtype. This can be seen distinctly in PR, H2AX, and PCNA. The predictions receive poor correlations for all subtypes when the marker does not show substantial positive expression, and the predictions receive good correlations whenever the subtype does show a variable expression range. Although many of the low correlation scores around 0.70 are still adequate, their reduced performance compared to the other markers is due to their low variability in a breast cancer subtype. This can be further seen in PCNA where the correlation metric is 0.39 when the marker is completely absent from the subtype. This absence of markers skews the evaluation of the models. This also further illustrates the

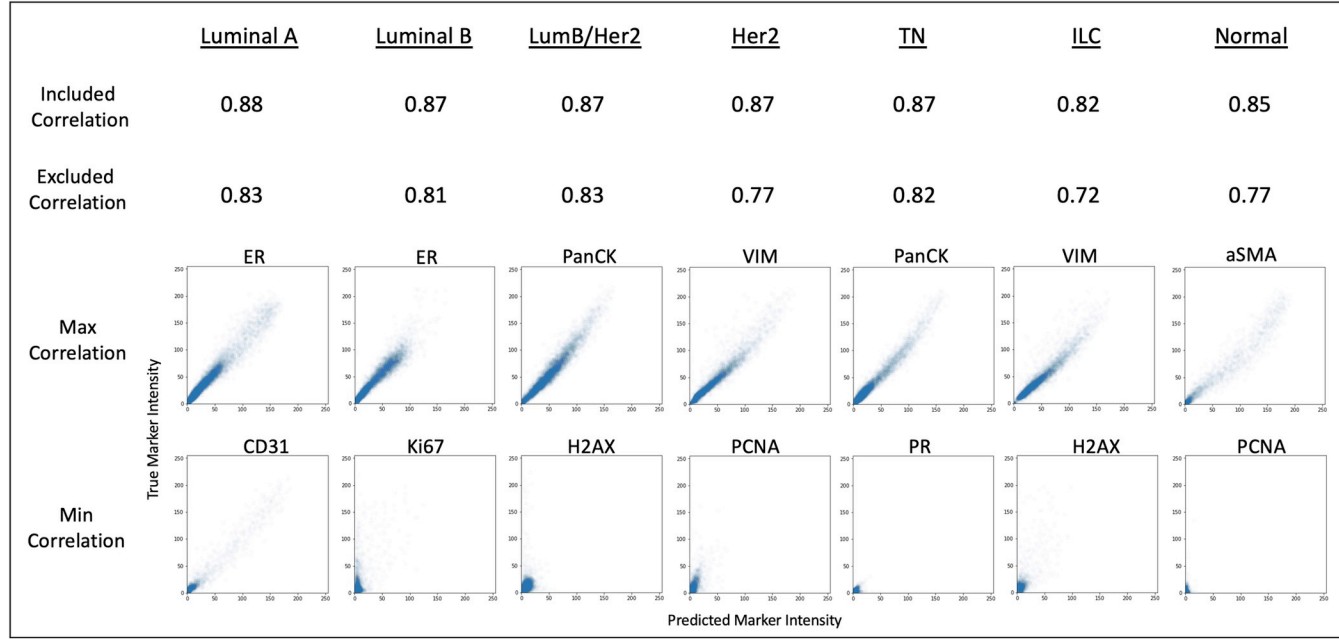

**Fig 4. The full panels of six different breast cancer subtypes and normal were predicted using the highest performing reduced panel (correlation-based selection with 18 markers).** Spearman correlations were calculated between the full panel expressions and the expressions of the predicted markers for the included markers and excluded markers separately. The expression correlation plots for the best and worst predicted markers are shown for each subtype.

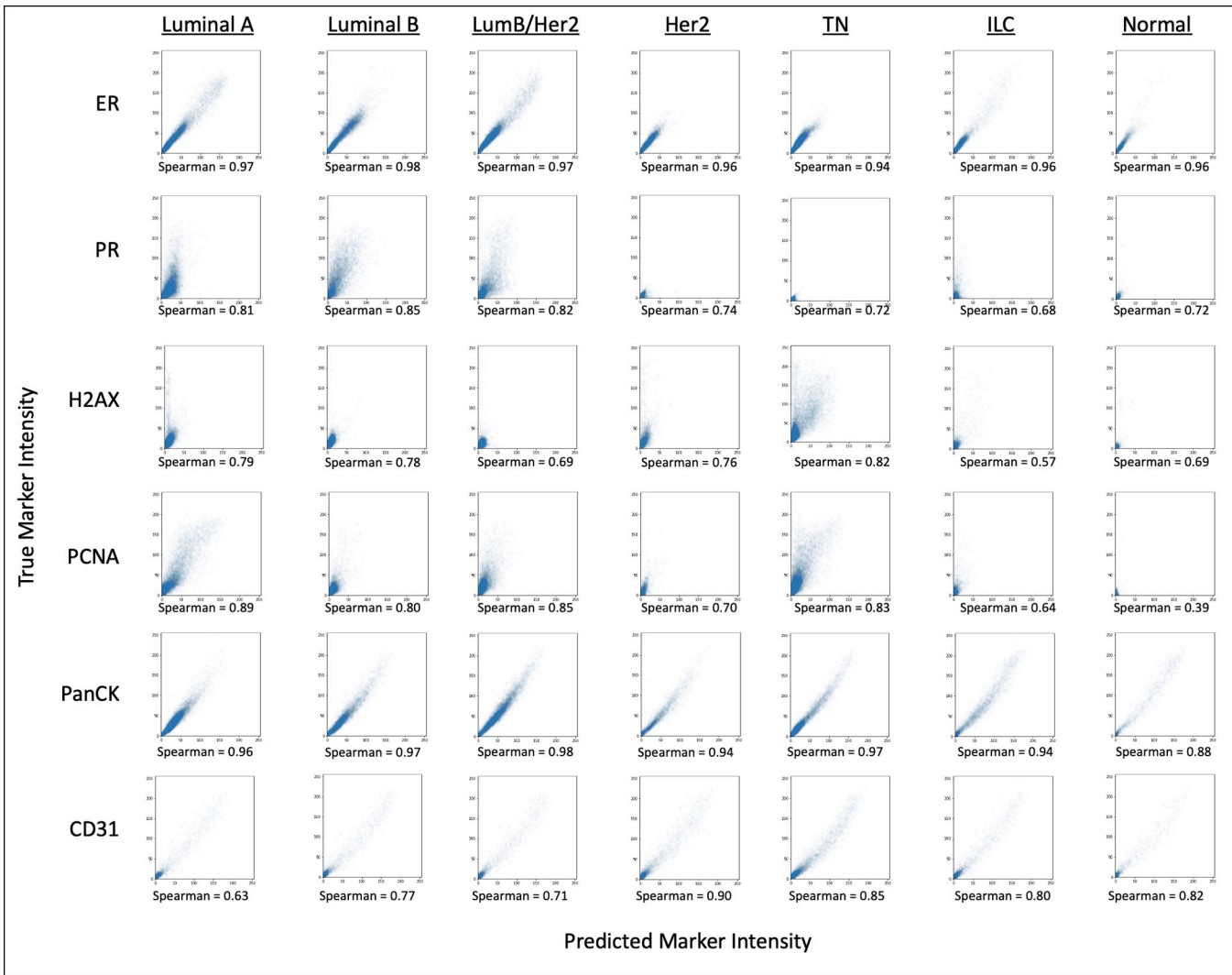

**Fig 5. A sample of a few of the lowest scoring and highest scoring makers were selected to directly compare the Spearman correlations across all the breast cancer subtypes.** True and predicted expressions were compared for each marker and subtype individually.

consistency of the panel across subtypes despite the differences in biology and marker expression, since the predictions only score low correlations for absent and low variability samples. Portions of all patient samples were used during training due to small dataset size, so generalizability here is only with respect to breast cancer subtypes.

## Evaluating selection methods with cluster matching

Although it is important to be able to reconstruct the mean intensities of single cells, downstream analysis such as single cell phenotyping and clustering is important for biological research, and if such analytical methods were to be affected, then the reduced panel predictions would not be useful for complex research methods. As shown in Fig 6, although the selection methods have varied levels of performance at predicting mean intensity, when 18 of 25 markers are included in the reduced panel sets, all selection methods perform well at recapturing the same clusters extracted from the full panel set, as measured by normalized mutual

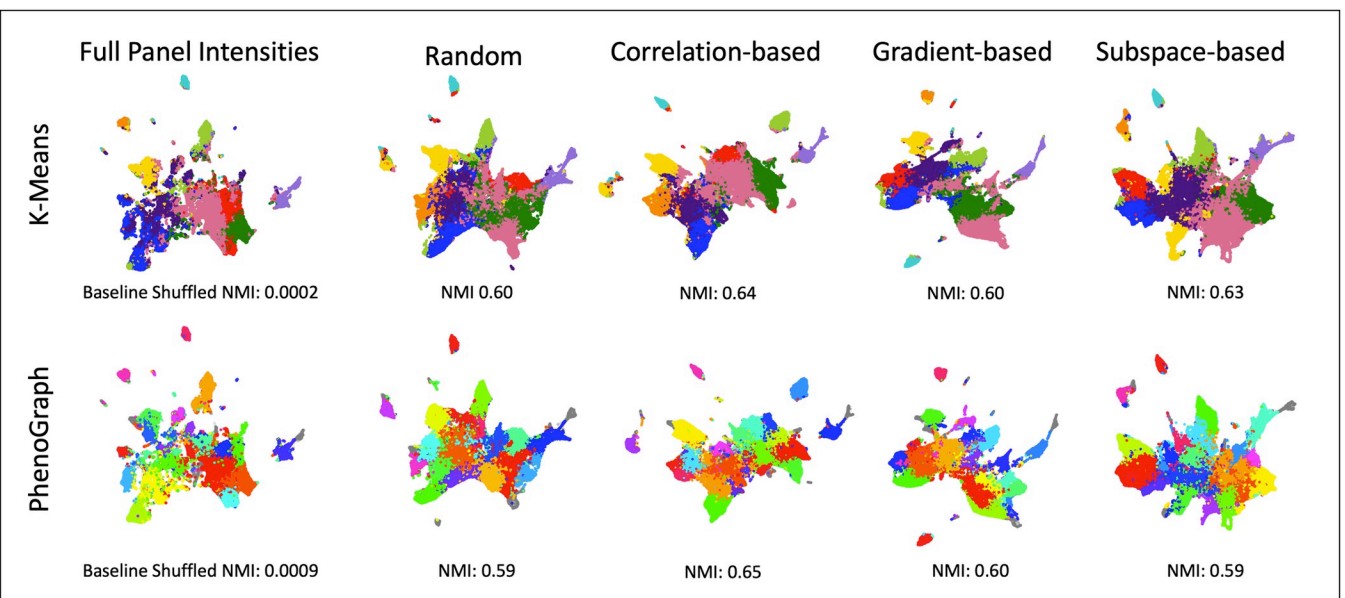

**Fig 6. Clustering was performed on the full panel intensities to generate ground truth cell type clusters using k-means (k = 10, chosen with elbow method on silhouette score) and PhenoGraph[22] (with nearest neighbors set to 500 and minimum cluster size set to 2000).** Random, correlation-based, gradient-based, and subspace-based selection methods were also clustered using reconstructed intensities as input to k-means and PhenoGraph using the same parameters. Clustering similarity to ground truth was performed using normalized mutual information (NMI). A baseline NMI for comparison was generated using randomly shuffled cluster labels. The clusters were projected into a UMAP[23] embedding and plotted to visually show the cluster results. For k-means clustering, cluster colors were matched for all selection methods by pairing each cluster's cell compositions with the full panel. This was not done in PhenoGraph because each method had a different number of clusters, preventing 1-to-1 pairing. Outliers from PhenoGraph clustering are shown in grey.

information (NMI)[9]:

$$NMI\,(U,V) = \frac{MI\,(U,V)}{mean\,(H(U),H(V))} \qquad (1)$$

where U and V are the reduced panel predicted and full panel (ground truth) cluster labels and H(U) and H(V) represent the entropy of U and V, respectively. The predicted clusters were then paired to their full panel counterpart by examining the population compositions to maximize consistency.

This again shows that the information within the 7 withheld markers is able to be predicted using the 18 markers in the reduced panel, enough to produce similar downstream results for clustering and potentially cell phenotyping. Although it would be ideal to compare results to ground truth cell types, the dataset was limited by lack of labeled cell type information; therefore, the clustering results from the full panel mean intensity dataset were used as ground truth for the single cell populations that can be extracted. Using k-means as the clustering method (with n = 10 selected using silhouette score), the correlation-based method achieved the highest NMI of 0.64, while gradient-based, subspace-based, and random selection achieved only slightly lower NMIs of 0.60, 0.63, and 0.60, respectively. All k-means clustering results are significantly larger than the baseline NMI of randomly shuffled cluster labels (NMI = 0.0002). Using PhenoGraph[22] as the clustering method (with nearest neighbors set to 500 and minimum cluster size set to 2000), the correlation-based selection method achieved the highest NMI of 0.65, while the gradient-based, subspace-based, and random selection methods achieved lower NMIs of 0.60, 0.59, and 0.59, respectively. All PhenoGraph clustering results are significantly larger than the baseline NMI of randomly shuffled cluster labels (NMI = 0.0009).

Although the spatial distance in the illustrated Uniform Manifold Approximation and Projection (UMAP) [23] cluster plots is not quantitative in regards to similarity of clusters, one can qualitatively see the same overall pattern and organization of clusters between the full panel and all selection methods. This shows that while intelligent selection of the reduced panel will matter for some forms of downstream analysis such as mean intensity metrics, the selection method might be irrelevant to the end result of other analytics such as clustering since the deep learning architectures can learn to capture the most defining information of a full panel so long as they receive enough information volume during training.

## Discussion

Despite the increased panel capacity of many novel multiplex imaging modalities, it is important to select the most biologically relevant markers in order to save resource, time and effort, and to reduce the amount of negative tissue effects such as tissue degradation or tissue loss in later rounds of staining as well as to add a room for other biologically important markers. Because of the complex inter-connectedness of cellular biology within breast cancer TMAs, many markers are self-correlated and do not add significant information that is not already conveyed by other markers on the panel. Here we present that by using marker correlation to select an optimal panel, we can reduce the panel size by 72% while retaining expression information with a 0.90 correlation (Fig 3), and recapitulate downstream clustering results (Fig 6) within a CyCIF breast cancer TMA dataset.

The integration of computational panel design into existing multiplex imaging pipelines will enable researchers to design more efficient panels, maximizing the biological findings while using less time, effort, and resources to produce their images. Furthermore, by removing easily predicted markers from their panels, researchers will be able to introduce new markers that they couldn't include before, allowing them to capture new information that is not redundant with the other markers already on the panel. Although there will always be the potential for information loss when removing markers and it cannot be known which markers will have unique expressions in a sample ahead of time, this methodology can help guide researchers to make decisions that will capture the most relevant data from the samples with less reliance on unquantifiable decisions.

The limitation of all these methods of panel selection is that they require a round of staining to be conducted first so that the marker interactions can be measured and evaluated to determine the level of panel self-correlation. After an initial pilot panel, these optimally reduced panels would be able to improve experiment time and expense by decreasing the number of staining rounds required. Ideally, it would be best to use the information gained from these selection methods to design panels for new datasets without having to stain, test, and design for every new dataset, tissue, or patient. The selected subpanels, however, would only be applicable within the context of the same full panel and biological context, and may not extend properly outside of that setting. This work also does not reflect the capacity to extend findings to new patient data, as portions of data from all patients were used during training. Furthermore, the information retained after withheld stain prediction might only be with respect to the metrics used for evaluation and biases of the architecture used for imputation. There is likely information not tested for that is being lost through imputation even when scores show good results by certain metrics and there is information that one sampling method will retain that another method will not, even if the architecture is unable to reconstruct that information due to the inherit inductive biases in reconstruction method. The findings shown here are not necessarily a general evaluation of stain overlap or reducibility, but are only an evaluation with respect to the detailed metrics for a single imputation architecture.

Single cell stain prediction has been performed previously [24]; however, this work only used one segmentation method to create single cell images, so additional research can be conducted to determine how well prediction and reduction methods generalize to various segmentation qualities. Similarly, only one architecture for unseen stain imputation was utilized here. It is possible that different architectures and methods for stain prediction will favor different sampling and reduction methods. We recommend further research to determine the degree to which this effects results. Although we demonstrate this method's utility for identifying reducible markers within a single diverse dataset of BC subtype as a proof-of-concept study, to the best of our knowledge, this is the first study for intelligent marker selection and evaluation of several marker selection methods in the multiplex imaging setting. Future research can look into the deployment of the designed panels to new datasets, disease states, and patient samples without the need for retraining. Research can also be done into the biological relevance of the reduced sets so that researchers can better design panels on their own with fewer excess predictable markers. By identifying which markers are consistently well predicted and which consistently fail regardless of panel reduction method, researchers can design future panels with informed decisions to include the poorly performing markers and exclude the easily predicted markers.

## Methods

### Panel reduction dataset

The dataset used for testing panel reduction methodologies was a breast cancer tissue microarray (TMA) available on synapse from the human tumor atlas network (HTAN) TNP-TMA [24]. As part of this paper all images at full resolution and all derived image data (e.g. segmentation masks) will be publicly released via the NCI-recognized repository for Human Tumor Atlas Network (HTAN; https://humantumoratlas.org/) at Sage Synapse. A version of this data is available at https://www.synapse.org/#!Synapse:syn22041595.

The BC TMA dataset is comprised of 88 cores and 6 different cancer subtypes: luminal A, luminal B, luminalB/HER2+, HER2+, Triple Negative, and Invasive Lobular Carcinoma. Reference tissue, normal breast, and cell lines are also included. The BC TMA was imaged using cyclic immunofluorescence with 40 marker channels. The imaging channels were filtered down to 25 channels of interest by removing autofluorescent and duplicate marker channels such as nuclear staining (see S1 Table). The stained images were normalized using histogram stretching to the $1^{st}$ and $99^{th}$ percentiles, ignoring background area which was thresholded manually. Cell masks were generated by UnMicst/S3seg using MCMICRO pipeline [25,26] where the cell ring mask was dilated 3 pixels from the nuclear mask. The segmentation resulted in a total of 737,653 single cells images. As described in [9], transformational features of single cell images can skew the latent spaces of encoding models like a variational autoencoder (VAE). For this reason, the single cell images were corrected by rotating all images such that the major axes of all cell masks were aligned and were corrected for polar orientation by flipping the images such that the center of staining mass was located in the same quadrant for all cells. By doing this, the model can focus on relevant staining information and ignore transformation information that is irrelevant to retaining panel information. All selection methods listed in the following sections were performed using the correlations, interactions, and intensities of the full dataset. The image reconstruction model was trained using a randomly selected 90% of the single cell images. The other 10% of the single cell images were used for quantitative and comparative analysis of the selection methods, i.e. reconstruction accuracy, intensity correlation, subtype generalizability, and downstream clustering (Figs 2, 3, 5 and 6).

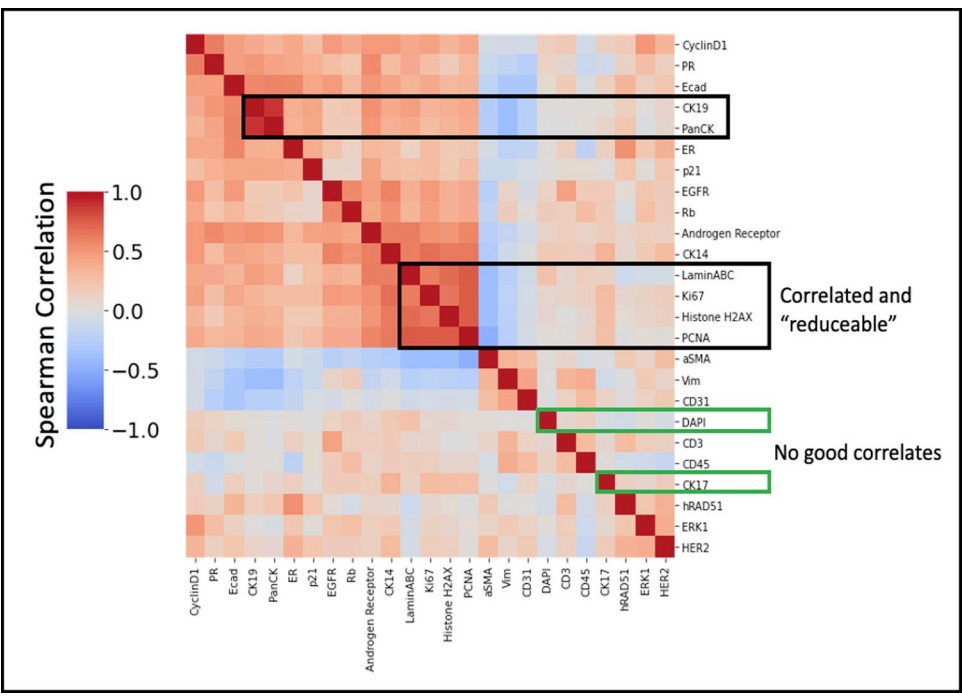

**Fig 7. Heatmap of mean marker intensity correlations in the full TMA panel set, computed across single cell images.** Heatmap visualization is clustered using hierarchical clustering of rows and columns. Highly correlated marker clusters show where markers can potentially predict one another and thus can be reduced. Markers with no good correlates will likely need to be included in a reduced panel as there will be no other marker that is predictive of their expression (using intensity information alone). Baseline 1-to-1 substitution will use these correlations to determine marker pairs for intensity substitution. Correlation-based selection will combinatorially create and test all possible panels of size *n* to determine which reduced panel produces the max correlation to all withheld markers.

## Methodologies for selecting the optimal reduced panel

Within a set of markers, intensity information is often correlated when a portion of the proteins of interest operate along the same pathways, are mutually expressed, or are tied to similar phenotypic states. This can be true for markers that localize to different regions of the cell so long as they are correlated in overall expression for different cell states (Fig 7). Although some of these correlated stains might be selected for biologically relevant reasons, quantitatively the information from one or more markers can be used to predict the information of another, meaning that they can be reduced. Based on this, there exists an optimally reduced panel that maximizes the amount of information gained using the fewest markers while preserving all information.

## Baseline 1-to-1 intensity substitution using reduced panel only

In order to create a baseline comparison of reduced panel performance, it is necessary to access the maximum amount of information retained using just a reduced panel without computational inference. Since the metric for evaluation is the correlation between predicted and ground truth marker expression it was necessary to create predictions of withheld markers from the reduced panel without computation. A simple 1-to-1 expression substitution was used for baseline because it is the least computationally intensive method. To do this we simply computed the correlation for each marker within the full panel set and paired each withheld marker to its highest correlated partner as shown in Fig 7 within the randomly selected

reduced panel set of a given size. To simulate the predictive inference that would be made by only having the reduced panel set and no reconstruction methods, the predicted intensity for each withheld stain was simply the intensity of its matched partner. This produced fairly low correlations for all panel sizes that only converged to 1 when nearly all the stains were included in the reduced panel (Fig 3).

### Random selection

In order to determine the importance of intelligently selecting panels, we simply generated random panels for each panel size to be used as a baseline panel selection comparison. For ease of testing, we generated a random sequence of markers, and the markers were sequentially added to the panel in that order. The list of markers used for each randomly selected panel can be seen in S2 Table.

### Intensity correlation-based panel selection

In order to determine an optimally reduced panel, the stains that maximized the correlation to all the stains withheld from the panel were chosen. If the correlations between all the stains in the dataset are pre-computed (Fig 7), one can quickly perform a combinatorial test of all possible reduced panels with $n$ markers. For every combinatorially created potential panel, the withheld markers are paired with their highest correlated marker within the potential panel, and the max correlations for all withheld markers are averaged to assign a score to that potential panel:

$$Panel\ Score_{corr} = \frac{\sum max\ [corr(W_i, R)]}{length(W)} \tag{2}$$

where $R$ is the potential reduced panel being evaluated, $W$ is the set of withheld markers, and $W_i$ is the intensities of each withheld marker that is being paired. Once we have scored every potential panel of a given size, we then select the panel that has the greatest score, indicating its predictive capacity toward the withheld markers. Although this method is simplistic, utilizing only mean intensity information, it is quick and is not computationally intensive, making it amenable to rapid panel design and testing. Using this method, the markers are re-selected for each panel size, meaning a specific marker might be included in a panel of one size but not in the next. It is worth noting that for all selection methods, the nuclear staining marker was a requirement for inclusion in the panel since it is a common marker among currently implemented panels and is a necessary marker for most segmentation pipelines. Also, as Hoechst visualizes nuclear DNA content and it has been used to assess cell morphology or cell cycle phase estimation, it is a potentially important marker to predict other marker expression in the reduced panel setting. The panels selected using this method can be seen in S4 Table.

### Sparse subspace-based panel selection

Although the correlation-based method is quick, simple correlation of mean intensities ignores potentially more complex interactions between markers, as two or more markers might be required to predict the expression of a third. Also, combinatorial testing, while quick on small numbers of stains, can become exponentially more burdensome to compute with large panel sets. Inspired by sparse subspace clustering [27] or using self-expressiveness property, the next method tested seeks to detect complex interactions across the panel set. The key idea is that, among many possible representations of a single cell marker expression in terms of other markers, a sparse representation corresponds to selecting a few marker expressions from the same subspace. To do this we train a coefficient matrix ($C$) such that the matrix multiplication

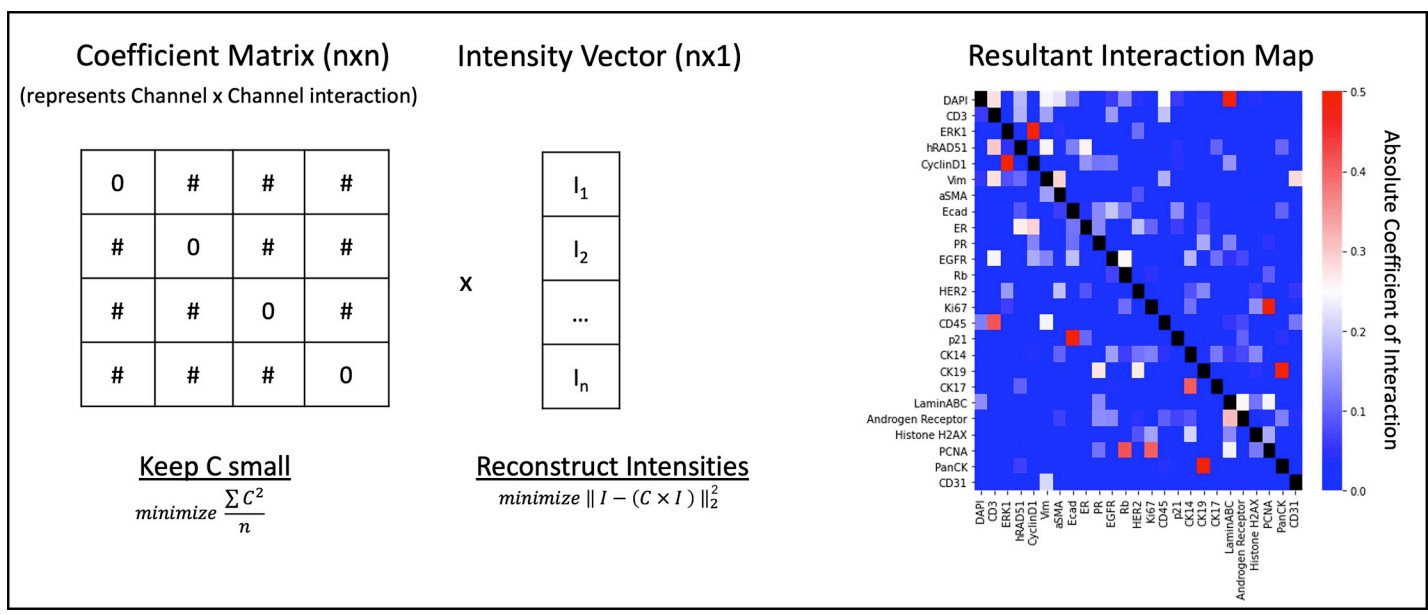

**Fig 8. Diagram demonstrating the trained coefficient matrix and the resultant interaction map used to select a reduced panel.** A model is trained to optimize the Coefficient Matrix (*C*) with a forced zero diagonal, such that it is sparse and when multiplied by the intensity vector of each single cell (*I*) it can reconstruct *I* as closely as possible. The resultant interaction map is the trained weights of *C*, showing the interactions of each marker necessary to adequately reconstruct each other marker in an image. Some makers are capable of being reconstructed from only one other marker, other markers require a more complex combination, and some are not well predicted by any.

of *C* and the single cell marker-wise intensity vector (*I*) reconstructs *I* as accurately as possible (Fig 8). Here *I* is an $n \times 1$ matrix where *n* is the number of markers in the full panel set. During training the diagonal of *C* is forced to be 0 so that the matrix does not converge to the identity matrix and the off-diagonal coefficients ($C_{ij}$) give information of the interactions necessary for reconstruction. Training of the *C* matrix uses the following loss:

$$L_C = \frac{\sum C^2}{n} + \left\| I - (C \times I) \right\|_2^2 \qquad (3)$$

where $C_{ii} = 0$, $i = \{1, \cdots, n\}$. Before clustering and analysis all interactions with a value below a threshold (here $< 0.05$) are dropped. This process penalizes the non-zero element of *C* such that 1) it remains sparse and only places weights on a few interactions that contribute the most to reconstruction of the intensity vector and 2) penalizes the accuracy of *I* reconstruction such that the model learns to compute accurate and relevant interactions as shown in Fig 8 right. Looking at the resultant interaction map, one can see that many markers can play a part in predicting the relative intensity of another marker. To determine an optimal panel set from the interaction map, a similar combinatorial method was used, similar to the correlation-based method. Here, however, we attempted to select a theoretical reduced panel that maximized the interactions to the withheld markers, while minimizing the interactions within the panel:

$$Panel\ Score_{SSC} = \frac{\sum Int(W_i, R)}{length(W)} - \frac{\sum Int\left((R_i, R)\right]}{length(R)} \qquad (4)$$

where *R* is the potential reduced panel being evaluated, *W* is the set of withheld markers, $W_i$ is the interactions of each withheld marker to the markers in panel *R*, and $R_i$ is the interactions of each included marker to the other markers in the reduced panel. Once we have scored every potential panel of a given size, we then select the panel that has the greatest score, indicating its

predictive capacity toward the withheld markers. Just like with correlation-based selection, the reduced set of markers is re-selected for each panel size, meaning a specific marker might be included in a panel of one size but not in the next. The reduced panels selected using this method can be found in S5 Table.

### Deep learning gradient-based selection

While both of the previous methods rely solely on mean intensity information, image data has significantly more information than intensity readouts alone. The localization, texture, and shape of cells can also tell us about potential co-staining patterns and interactions. In order to quantify how these features may be important, we use a deep learning method that quantifies the channel-wise importance for reconstructing imaging features across all channels. A similar method to the one described here uses the gradient of the model to determine the channel-wise importance for cell type classification [28]. The key difference of our proposed method is the objective of the model (reconstruction instead of classification) which requires a different architecture. Instead of using a series of ResNet encoders and fully connected classification layers, we use a series of encoders equal to the number of input channels and a decoder for the purpose of reconstructing the combined input channels (Fig 9). This forces the gradient at the encoding layer to be greater for channels that play a bigger role in the reconstruction of the full panel image, including localization, textures, and intensity information. Since the encodings of each channel are kept separate and concatenated, the magnitude of the gradients can be averaged for each channel separately and evaluated for their importance. The gradient at the encoding layer is determined using the Tensorflow built-in method of GradientTape [29]. The reduced panel set is then selected by taking the top $n$ channels ranked by importance, where $n$ is the desired panel size. Because the importance is static and can be sequentially added in order or ranking, the selected panels have the advantage of simply being expansions of the smaller panels, potentially making panel design easier. The list of ranked markers can be found in S6 Table.

### Model architecture for imputing full images and calculating gradient

In order to impute the full panel image from the reduced panel, we trained a multi-encoder variational autoencoder (ME-VAE) [9] where the inputs to each encoder were the channels of

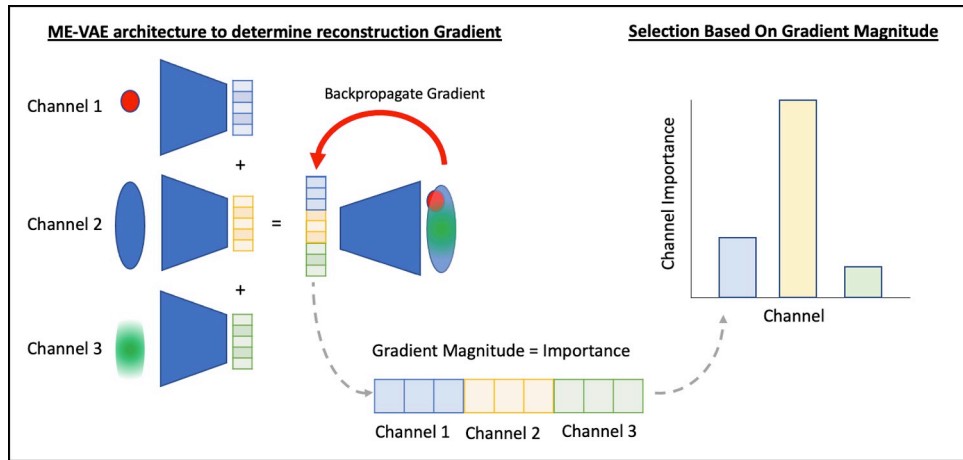

**Fig 9. A multi encoder variational autoencoder architecture is implemented with each channel being used as the input to parallel encoders.** The encodings of each channel are concatenated and decoded into a full panel image. The gradients of the model are then backpropagated to the encoding layer. If the magnitude of the gradient is interpreted as importance, the channel gradients can be averaged across the dataset to determine which markers are most important for reconstructing image features within the model.

the input set and the output was the full panel image. The encodings from each input were concatenated into a single vector before being passed to the decoder. Each encoder and decoder network has 3 layers deep, and each layer used a rectified linear unit activation, except for the output layer which used a sigmoid activation. The concatenated encoding dimension was kept to $\sim$ 128 for all reduced panel sizes. Each input is encoded into its own latent space, equivalent in length to the latent space of all other inputs, meaning that it was not always possible to get a total concatenated latent space of 128 depending on panel size. For this reason a total latent space as close to 128 was chosen for each panel size. Gradient calculation was performed using the same model, where the input was the reduced panel channels, the output was the full panel image, and each channel's encoding was kept separate, allowing for evaluation of gradient and the per channel level. These models use a modified ME-VAE loss:

$$L_{ME-VAE} = BCE(x, p(z_{all})) - \frac{1}{n}\left(\sum_1^n KL[q_i(z_i|(x_i)||p(z_{all})]\right) \tag{5}$$

where each encoder's ($q_i$) individual latent space ($z_i$) is combined in a concatenation layer to create a mutual latent space ($z_{all}$), $x_i$ represents a different channel of image $x$, and $n$ represents the number of markers in the reduced panel. Using the described setup, models were trained for 10 epochs and 90% of the dataset.

## Metrics for reduced panel evaluation

Three metrics were used in the evaluation of reduced panels. In order to evaluate the mean marker intensity predictions, Spearman correlation was used comparing each marker's mean intensity between ground truth and reconstructed images. The Spearman correlation was computed for each stain individually, and then the average correlation across all stains in the set was reported. Correlation variance was also computed across stains to determine whether stain prediction was consistent i.e., predicting all stains reasonably well instead of just some stains extremely well and others not. In order to evaluate the quality of the reconstructed CyCIF image, we also computed the Structural Similarity Index Measure (SSIM) [21] between each single cell image and its reconstruction, which is a standard metric for tasks like image translation, image denoising, and image restoration because it evaluates both the intensity and spatial information of the image. SSIM was implemented in python using the sci-kit learn package [30]. This was done for each channel individually, and then the average across all cells and channels was reported. In order to evaluate the retention of information necessary for downstream phenotyping, Normalized Mutual Information (NMI) was computed for the cluster labels created from the full panel intensities and for the reconstructed intensities of each selection method. NMI was implemented in python using the sci-kit learn package [30]. For each selection method, the panel size used for this test was 18. For full panel and all selection methods, k-means was used for clustering with k set to 10 (determined using the elbow method on silhouette score for the full panel dataset). K-means and silhouette score were computed in python using the sci-kit learn package [30]. UMAP embeddings, calculated from reconstructed intensities, were also used to visualize the clustered data using the UMAP package [23]. Cluster labels were colored such that the reduced panel clusters matched to the cell composition of the full panel clusters. PhenoGraph [22] was also used to create clusters for evaluation of each method, with nearest neighbors set to 500 and minimum cluster size set to 2000. PhenoGraph cluster plots could not be paired 1-to-1 for color matching on plots since each selection method produced a different number of clusters. Outliers from PhenoGraph analysis are shown in grey.

### Methods for simulating technical noise

In order to frame the accuracy of predictions in a biological context (S1 Fig), we simulated several types of technical noise commonly found in imaging data (blurring, salt/pepper, and variation in segmentation method). To simulate image blurring, we used the scikit-image implementation of gaussian blur [30] with sigma set to 1. To simulate salt/pepper noise, we used the scikit-image implementation of random_noise with the mode set to "s&p" and the amount set to 0.1 [30], which created a moderate amount of salt and pepper noise in the image. To simulate variation in segmentation, we applied erosions and dilations to image masks, eroding half the image and dilating the other half. The half of the image that was to be eroded/dilated was chosen arbitrarily and kept consistent across all single cell images. This created a mask deformed from the original by a few pixels, as can reasonably be expected from different segmentation methods. Images were then re-extracted using the new masks.

## Supporting information

**S1 Fig. To frame the extent of error in the predicted results from a randomly selected reduced panel of 12 stains, several technical noises were simulated and evaluated for the same metrics.** The average SSIM was measured for each stain individually and averaged. Likewise, the Spearman correlation between the original stain intensity and the resultant stain intensity was calculated for each stain independently and averaged across the withheld panel set.
(PDF)

**S1 Table. A list of full breast cancer TMA panel marker set.**
(CSV)

**S2 Table. Randomly selected reduced panel set.**
(CSV)

**S3 Table. Metrics of model comparison.**
(CSV)

**S4 Table. Correlation-based reduced panel set.**
(CSV)

**S5 Table. Sparse subspace-based reduced panel set.**
(CSV)

**S6 Table. Gradient-based reduced panel set.**
(CSV)

## Acknowledgments

We thank Drs. Peter K. Sorger, Guillaume Thibault, Laura Heiser, Daniel Zuckerman, and Gordon Mills for providing useful feedback. The resources of the Exacloud high performance computing environment developed jointly by OHSU and Intel and the technical support of the OHSU Advanced Computing Center are gratefully acknowledged.

## Additional information

Accession codes code available from the author's Github and tutorials (https://github.com/GelatinFrogs/ME-VAE_Architecture and https://github.com/GelatinFrogs/PanelSelectionMethods);

## Author Contributions

**Conceptualization:** Joe W. Gray, Young Hwan Chang.

**Data curation:** Jia-Ren Lin, Yu-An Chen.

**Formal analysis:** Luke Ternes.

**Funding acquisition:** Joe W. Gray, Young Hwan Chang.

**Methodology:** Luke Ternes.

**Project administration:** Young Hwan Chang.

**Software:** Luke Ternes.

**Supervision:** Young Hwan Chang.

**Writing – original draft:** Luke Ternes, Young Hwan Chang.

**Writing – review & editing:** Young Hwan Chang.

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
