## [Decision Letter · Decision Letter 0]

26 May 2022

Dear Dr. Chang,

Thank you very much for submitting your manuscript "Computational Multiplex Panel Reduction to Maximize Information Retention" for consideration at PLOS Computational Biology.

As with all papers reviewed by the journal, your manuscript was reviewed by members of the editorial board and by several independent reviewers. In light of the reviews (below this email), we would like to invite the resubmission of a significantly-revised version that takes into account the reviewers' comments.

We cannot make any decision about publication until we have seen the revised manuscript and your response to the reviewers' comments. Your revised manuscript is also likely to be sent to reviewers for further evaluation.

Sincerely,

Min Xu, PhD

Guest Editor

PLOS Computational Biology

Jian Ma, PhD

Deputy Editor

PLOS Computational Biology

Reviewer's Responses to Questions

**Comments to the Authors:**

Reviewer #1: Review: Computational Multiplex Panel Reduction to Maximize Information Retention

The authors propose to address a very interesting and impactful problem, of how to select a reduced set of stains in multiplexed images, that capture a similar amount of information as a larger panel. Their main claim is that, out of several strategies, using a simple metric of correlation between stains to determine which stains to withhold performs well and generalizes to unseen cell lines within breast cancer cells. However, I think there is one fatal flaw in the experimenting supporting this claim: because each of the withholding strategies results in different stains being predicted by their models, and the metrics are reported on reconstructing the particular stains withheld from a model, the metrics between models in this paper are not comparable. This is because different stains can be more or less difficult to predict: more spatially variable stains are more challenging than homogeneously distributed stains, and from my reading, there has been no attempt to fairly compare models where the more challenging stains are given as training inputs versus being held out as test inputs.

A second major flaw is that the authors claim that they can choose stains that maximize the "information" retained in this paper, but as far as I can tell, the evidence supporting this is weak. The only attempt at showing that biologically relevant information is retained is Figure 6, which shows that clustering mean intensity (I think, the writing is somewhat ambiguous) from a reduced set of stains generally has around 0.60 normalized mutual information with the full panel of stains. However, this experiment is weak for multiple reasons: there is no evidence that clustering of single cell features in this context retrieves cell types; as far as I can tell, there is only a very limited attempt to extract features from the images (i.e. only intensity, not shape or distribution); the clustering method and parameterization is arbitrary and simple, and there appear to be subtypes of cells in the dimensionality reduction that the clustering method is not retrieving in the full panel if we even believe this metric biologically relevant; and there is no strong baseline to interpret what a 0.60 NMI even means in this context (only a very weak randomly shuffled baseline). Otherwise, the rest of the paper shows evidence that the distribution of stains can be predicted well in and out of training distribution, which is not the same as the titular claim.

There are other comments I have about the clarity of the paper, some technical points of the methodology, and the metrics used - I hope these will be helpful for the authors. But given that the experiments supporting the main claims of the paper are technically flawed, and only weakly support the titular claim of the paper at best, I have no choice but to recommend a reject unless the authors can demonstrate to me that I've missed something major in their work.

Major Comments:

1. The primary metric supporting most of the claims in this paper is the reconstruction quality of unseen stains, measured by the Spearman correlation and SSIM between the predicted outputs for these stains and their actual stains. This is used to support the claim that a simple metric of marker correlation performs best at selecting a maximally informative reduced subset of markers. I would like to posit an alternative hypothesis: the marker correlation metric is selecting stains to withhold that are most spatially homogeneous (i.e. more uniformly distributed across the cell/nucleus). These stains are systematically easier to predict, so the reason why we're seeing higher correlations is simply because some models have been given harder images to predict (e.g. more spatially heterogeneous images), while others have been given easier ones. I would like to see this ruled out by showing a baseline where the N channels being predicted are the most spatially homogeneous channels. More broadly, I would like to see the issue that some models are given easier prediction problems than others be addressed - the metrics across models are just not comparable currently. One solution would be to have a single standard set of held out markers, and then choose the subset along the remaining markers that best predicts these markers, instead of reporting a metric on variable outputs.

2. Is this paper about single cell phenotyping, or single nuclei? The segmentation method used seems to work on nuclear dyes exclusively, which suggests the latter, but the paper seems to suggest the former. This distinction is important and would impact the biological claims and limitations of the paper: reconstructing purely nuclear expression/localization is easier than reconstructing expression at the full cell level (which I expect would be much more difficult due to increased number of compartments and variability in cell morphology). It is not clear that a method built for imputation of stains in the nuclei exclusively would extend to full cells in general: the authors need to clarify this and better contextualize their method.

3. I'm confused about the evaluation in Figure 6, which seems to be very critically important given that this is the only experiment that attempts to validate whether the predicted stains have any utility to downstream biological analyses and not are just technically correct reconstructions of the stains. How are the authors obtaining the latent space for the full panel - is this just the average intensity of the cell per stain or do the authors have some kind of latent space for the full panel intensities? If it's the former, how does this relate to point 4? How was k for k-means selected? There's a lot of substructure in the full panel intensities (e.g. especially in Cluster 7) that doesn't seem to be resolved by any of the subsampled models - doesn't this suggest that there is information in the full panel being missed by the partial panels? In general, the arbitrary selection of clustering method and co-efficient, the limited features being employed here, and the lack of solid baselines for interpreting the NMI means that the claim that the deep learning architectures are capturing the "most defining" information of a full panel is fuzzy.

4. The segmentation procedure would interact with the experiments reported in this paper along multiple axes, but the paper doesn't go in enough detail about how exactly this was done. Greater clarity on the methods here would be appreciated. First, are stains used for segmentation also predicted outputs of the ME-VAE in various stratification set-ups? If so, how do the authors rule out that the stains are being predicted based upon the other stains, and not simply due to the segmentation mask? It seems that the segmentation mask would contain information about the localization of the stains, and potentially their intensity. Any stains used for segmentation should be excluded from the evaluation of the model. Second, how prone is this model to artifacts especially on this new, unseen data? My concern is that if the model is identifying artifacts or background as nuclei/cells, these kinds of mis-segmentations would tend to be bright or dim along all channels, so these would be easier to get.

5. As a practical point, do the authors consider spatial structure or intensity of stain to be the critical aspect to predict in these problems? My view is that the Spearman correlation is only going to capture the former, not the latter. But especially if this is nuclear expression, it seems like intensity is important for profiling here. Many of these stains appear to be transcription factors, where concentration in the nuclei is going to be important and not just spatial distribution. How well do the metrics reported in this paper align with the downstream profiling aims, given that it's possible to get the intensity of the marker totally wrong but still get a high correlation as long as the ranking of the pixels is preserved? (Also, is histogram stretching done per image, or globally using the percentile across all images? This seems critical as well.)

Reviewer #2: The authors develop a VAE to predict certain channels of multiplexed images from other channels. They apply several selection algorithms to identify the most predictive set of markers, and the set of markers than can be predicted. Their work can be used to guide panel design, which is a very important aspect in experiment design. The pipeline requires full panel data to identify the set of most predictive markers, suggesting the need for preliminary studies for each experiment, or use of prior knowledge. The manuscript establishes the feasibility of predicting certain markers from others, and establishes a data driven approach for marker selection. Future studies building on the current work can benefit panel design in the long run. Although I find the paper very interesting, there are a few concerns the authors need to address.

1- Given the study is done on one dataset, it is not clear if the correlations between true mean intensities and predictions are due to the power of deep learning models in extracting marker relations, or an artifact of panel design measuring heavily correlated markers in the first place. I would appreciate if the authors could comment if other multiplexed datasets show similar overall marker correlations, and if they believe similar correlations can generally be found in other datasets.

2- The authors need to better explain the train/test split in their methods section, e.g., if the train/test split is done at the cell level, image level, or patient level. The authors need to study how much the mean intensity prediction correlations may vary across patients, and how patient specific artifacts may affect the predictions of their model.

3- Sparse subspace-based panel selection method performs inferior to the simpler correlation-based panel selection method. I was wondering if the authors could comment why the subspace model performs inferior given that its assumptions on marker relations better conform to data properties.

4- I was wondering if the authors could explain their choice of cost function for subspace-based panel selection. The authors use the L2 norm of error ( || I-CI ||_2) instead of the conventional squared L2 norm (|| I-CI ||^2_2), and use L2 penalization followed by thresholding instead of a small L1 or elastic-net penalty to get a sparse solution.

5- The authors need to do more benchmarking of their VAE. Since the final mean intensity is the main objective, the authors need to compare their VAE with models that directly predict mean intensities. Despite the pixel level VAE can take advantage of subcellular spatial relations among markers, the extent the added complexity enhances prediction accuracy is not clear. This is important since in the marker selection step correlation based selection outperformed more complex algorithms.

6- I would appreciate if the authors could comment if they believe panel reduction generalizes to other multiplexed imaging techniques, such as IMC.

Reviewer #3: Review is uploaded as an attachment

**Have the authors made all data and (if applicable) computational code underlying the findings in their manuscript fully available?**

Reviewer #1: Yes

Reviewer #2: Yes

Reviewer #3: **No: **The authors present https://github.com/GelatinFrogs/ME-VAE_Architecture, but this repository does not contain the analysis code used in this paper from what I could tell.

PLOS authors have the option to publish the peer review history of their article (what does this mean?). If published, this will include your full peer review and any attached files.

Reviewer #1: No

Reviewer #2: No

Reviewer #3: **Yes: **Gregory P. Way
---

## [Decision Letter · Decision Letter 1]

9 Jul 2022

Dear Dr. Chang,

Thank you very much for submitting your manuscript "Computational Multiplex Panel Reduction to Maximize Information Retention in Breast Cancer Tissue Microarrays" for consideration at PLOS Computational Biology. As with all papers reviewed by the journal, your manuscript was reviewed by members of the editorial board and by several independent reviewers. The reviewers appreciated the attention to an important topic. Based on the reviews, we are likely to accept this manuscript for publication, providing that you modify the manuscript according to the review recommendations.

Sincerely,

Min Xu, Phd

Guest Editor

PLOS Computational Biology

Jian Ma

Deputy Editor

PLOS Computational Biology

[LINK]

Reviewer's Responses to Questions

**Comments to the Authors:**

Reviewer #1: Overall, the authors' comments have improved the clarity of their work: I now better appreciate that the authors trying to describe a method to select a reduced subset of stains agnostic of method, rather than reporting general insights from their own ME-VAE model. This helps address many of the concerns I raised - for example, the reason I emphasized spatial homogeneity is because this relates to the pixel-based loss of the model they used (fluorescence distributed evenly across the cell is easier to reconstruct with this loss, which has an "averaging" property on the image, than more "spiky" fluorescence patterns such as those distributed in punctuate compartments), but adversarial losses could easily overcome this restriction. Similarly, the reason I emphasized preprocessing is that it is not clear if these insights would generalize from images where the nuclei is identified and dilated versus an actual more accurate cell segmentation.

However, I still think that the authors need to go further in emphasizing these limitations in their writing, because there is the danger of readers still reading this paper and interpreting the insight that correlated-based selection is generally an effective strategic agnostic of methods. This is not tested, and cannot be concluded from this paper, and where I was originally misled. I appreciate how the authors have discussed the experimental limitation of their work (i.e. you do need a pilot staining where all the stains are produced first in order to be able to decide on the subset), but I think the authors should further discuss the technical limitation that what selection methods are effective are going to be dependent on the intended stain imputation method. Again, there is reason to believe the reason why a correlation-based method has done relatively well here could be because of the loss function used, and this may not be a general result over different loss functions.

Similarly, I still take issue with how the authors are framing their panel selection method as one that allows users to "maximimize the amount of information gained" in general, as opposed to in context of a specific stain prediction method. Stains that are difficult for one method to predict may be easy for another method to predict: in other words, this is not a method that allows users to reduce stains that share information with other stains in general, but a method that reduces stains that share information with other stains *to the extent in which the empirical method used can reconstruct this information, reflecting the inductive biases and limitations of the model*.

Reviewer #2: The authors have greatly improved the manuscript. I only have one remaining concern regarding the train/test split.

The authors mention train/test split is done at the cellular image level, suggesting some cells of a patient’s image are in the train set, and some are in the test set. This is concerning as patient ID becomes a confounder. Therefore, the results of Figure 5 are not reliable. To show the network can use biologically relevant information and not data artifacts, it is necessary to split data at the patient level.

Reviewer #3: The authors have adequately responded to all my concerns.

The authors uploaded a new github repository: https://github.com/GelatinFrogs/PanelSelectionMethods which is a good first step in ensuring reproducibility. I should have been more specific in my original review comment, which stated:

"The authors should make their analysis code publicly available. The authors present

https://github.com/GelatinFrogs/MEVAE_Architecture, but this

repository does not contain the analysis code used in this paper from

what I could tell."

The updated repository contains the panel selection code, but no implementation instructions. The authors should also archive their github code using a service like zenodo to ensure that the citable code exists elsewhere beyond the github repository.

**Have the authors made all data and (if applicable) computational code underlying the findings in their manuscript fully available?**

Reviewer #1: Yes

Reviewer #2: Yes

Reviewer #3: Yes

PLOS authors have the option to publish the peer review history of their article (what does this mean?). If published, this will include your full peer review and any attached files.

Reviewer #1: No

Reviewer #2: No

Reviewer #3: **Yes: **Gregory P. Way

Figure Files:

Data Requirements:

Reproducibility:

References:

---

## [Decision Letter · Decision Letter 2]

21 Aug 2022

Dear Dr. Chang,

We are pleased to inform you that your manuscript 'Computational Multiplex Panel Reduction to Maximize Information Retention in Breast Cancer Tissue Microarrays' has been provisionally accepted for publication in PLOS Computational Biology.

Best regards,

Min Xu, Phd

Guest Editor

PLOS Computational Biology

Jian Ma

Section Editor

PLOS Computational Biology

Reviewer's Responses to Questions

**Comments to the Authors:**

Reviewer #1: The authors have satisfied all of my comments.

Reviewer #2: The authors have adequately responded to all my concerns.

**Have the authors made all data and (if applicable) computational code underlying the findings in their manuscript fully available?**

Reviewer #1: Yes

Reviewer #2: None

PLOS authors have the option to publish the peer review history of their article (what does this mean?). If published, this will include your full peer review and any attached files.

Reviewer #1: No

Reviewer #2: **Yes: **Ali Foroughi pour

---

## [Editor Report · Acceptance letter]

26 Sep 2022

PCOMPBIOL-D-22-00517R2 

Computational Multiplex Panel Reduction to Maximize Information Retention in Breast Cancer Tissue Microarrays

Dear Dr Chang,

I am pleased to inform you that your manuscript has been formally accepted for publication in PLOS Computational Biology. Your manuscript is now with our production department and you will be notified of the publication date in due course.

With kind regards,

Zsofia Freund
